# Metabolomic Profiling of Cerebral Palsy Brain Tissue Reveals Novel Central Biomarkers and Biochemical Pathways Associated with the Disease: A Pilot Study

**DOI:** 10.3390/metabo9020027

**Published:** 2019-02-02

**Authors:** Zeynep Alpay Savasan, Ali Yilmaz, Zafer Ugur, Buket Aydas, Ray O. Bahado-Singh, Stewart F. Graham

**Affiliations:** 1Department of Obstetrics and Gynecology, Maternal Fetal Medicine Division, Beaumont Health System, 3811 W. 13 Mile Road, Royal Oak, MI 48073, USA; Ray.Bahado-Singh@beaumont.org; 2Oakland University-William Beaumont School of Medicine, Beaumont Health, 3811 W. 13 Mile Road, Royal Oak, MI 48073, USA; Stewart.Graham@beaumont.org; 3Beaumont Research Institute, Beaumont Health, 3811 W. 13 Mile Road, Royal Oak, MI 48073, USA; Ali.Yilmaz@beaumont.org (A.Y.); Zafer.Ugur@beaumont.org (Z.U.); 4Departments of Mathematics and Computer Sciences, Albion College, 611 E. Porter St., Albion, MI 49224, USA; baydas@albion.edu

**Keywords:** cerebral palsy, metabolomics, ^1^H NMR, targeted mass spectrometry, metabolic pathways, J0101

## Abstract

Cerebral palsy (CP) is one of the most common causes of motor disability in childhood, with complex and heterogeneous etiopathophysiology and clinical presentation. Understanding the metabolic processes associated with the disease may aid in the discovery of preventive measures and therapy. Tissue samples (caudate nucleus) were obtained from post-mortem CP cases (*n* = 9) and age- and gender-matched control subjects (*n* = 11). We employed a targeted metabolomics approach using both ^1^H NMR and direct injection liquid chromatography-tandem mass spectrometry (DI/LC-MS/MS). We accurately identified and quantified 55 metabolites using ^1^H NMR and 186 using DI/LC-MS/MS. Among the 222 detected metabolites, 27 showed significant concentration changes between CP cases and controls. Glycerophospholipids and urea were the most commonly selected metabolites used to develop predictive models capable of discriminating between CP and controls. Metabolomics enrichment analysis identified folate, propanoate, and androgen/estrogen metabolism as the top three significantly perturbed pathways. We report for the first time the metabolomic profiling of post-mortem brain tissue from patients who died from cerebral palsy. These findings could help to further investigate the complex etiopathophysiology of CP while identifying predictive, central biomarkers of CP.

## 1. Introduction

Cerebral palsy (CP) is the most common cause of severe neurodisability in children [1]. Although the main underlying causal factor is considered to be birth asphyxia, the pathophysiology of the disease is still not well understood. There are other causal factors that occur later in life that are hypothesized to be involved in the development of CP [2,3,4]. Congenital malformations are rarely identified [5]. Genetic predispositions with exposure to environmental factors can lead to CP. Common cerebral lesions seen in CP include destructive injuries, predominantly in the white matter in preterm infants and in the gray matter and the brainstem nuclei in full-term newborns [4]. The effect of these lesions, especially on the immature brain, could alter the series of developmental events [6]. Alteration in cell morphology or function and cell death observed in hypoxic ischemia or in inflammatory conditions leading to excessive production of proinflammatory cytokines [7,8], oxidative stress [9], maternal growth factor deprivation [10], extracellular matrix modifications [10], and excessive release of glutamate [11] have been shown to trigger the excitotoxic cascade and predispose the development of CP [12,13,14,15].

Cerebral palsy is a heterogeneous condition with multiple causes; clinical types and patterns of neuropathology on brain imaging; multiple associated developmental pathologies, such as intellectual disability, autism, epilepsy, and visual impairment; and, more recently, multiple rare pathogenic genetic mutations [2,16,17,18]. This is a clinical spectrum with many causal pathways and many types and degrees of disability [12]. These various pathways and etiologies have each resulted in a non-specific non-progressive disorder of posture and movement control. Thus, CP should be considered as a descriptive term for affected individuals, with each case requiring a detailed consideration of the underlying etiology. The described feature of this condition is one of the challenges for researchers due to the possibility of various underlying etiologies and confounders [12]. To our knowledge, there is currently no method available for predicting those at greatest risk of developing the disease. Moreover, only two strategies have succeeded in decreasing CP in 2-year-old children, which include the use of hypothermia in full-term newborns with moderate neonatal encephalopathy [19,20] and the administration of magnesium sulfate to mothers in preterm labor [21,22].

Among the new omics, metabolomics has the huge potential to advance our understanding of many complex diseases by uniquely detecting rapid biochemical pathway alterations and uncovering multiple biomarker panels, especially in various neurological disorders [23,24,25,26,27,28,29,30,31]. Over the past decade, the search for useful biomarkers to accurately predict brain pathology has become a growing area of interest. Biomarkers such as neuroimaging markers showing corticospinal tract integrity, metabolite ratios in brain regions, and brain volumes [32,33], multiorgan injury markers [34], and inflammatory markers [35,36] were studied as prediction models, however the description of metabolomic alterations or the identification of clinically approved biomarkers in CP has not been reported. There is accumulating evidence that metabolomic profiling of post-mortem brain tissue helps in understanding the pathophysiology of neurologic, neurodegenerative, and psychiatric disorders [23,24,28,37,38,39,40,41,42,43]. Thus, this study aims to biochemically profile post-mortem brain tissue from patients who died from CP and compare those with age-, and gender-matched controls. We believe that this approach will help us to identify central biomarkers of the disease while uncovering previously unreported biochemical pathways associated with the disease.

## 2. Results

Using ^1^H NMR and direct injection liquid chromatography-tandem mass spectrometry (DI/LC-MS/MS), we biochemically profiled post-mortem human brain tissue from people who died from CP and compared them with age- and gender-matched controls. We accurately identified and quantified 55 metabolites using ^1^H NMR and 186 using DI/LC-MS/MS. Figure 1 represents a labelled 1D ^1^H NMR spectrum acquired from an extract of caudate nucleus harvested from a person who died from CP. 

Due to the complementarity between the two techniques, there was a certain degree of observed overlap in terms of the metabolites measured (19 metabolites). To account for this, we took the average value for the individual metabolites and used this concentration value in our analyses, leaving us with 222 metabolites. Principal component analysis (PCA) was performed on the data to check for any intrinsic variation and subsequently remove any potential outliers (*p* < 0.05) based on the Hotelling’s T^2^ plot. No outliers were detected. Univariate analysis of the data revealed that of the 222 metabolites, 27 of them were at statistically, significantly different concentrations between CP and control tissue (Table 1; *p* < 0.05; *q* < 0.3). A full list of the 222 measured metabolites is available in Appendix A in the Appendix A.

Those compounds highlighted in bold are considered statistically, significantly different. (W)-data were non-normally distributed and the *p*-value was calculated by the Wilcoxon–Mann–Whitney test.

Having confirmed that there were significant differences between CP and control brains, we wanted to investigate a number of machine learning techniques to identify which method worked best for accurately discriminating between CP cases and controls. We used the variable importance functions *varimp* in h2o and *varImp* in caret R packages to rank the models’ features in each of the predictive algorithms. Feature predictors were estimated using a model-based approach. In other words, a feature was considered important if it contributed to the model performance. We extracted 20 important predictors from each of the models used for predicting CP. From these 20 features, the top metabolites were chosen and used to generate the specific predictive model. These were also compared across the different machine learning approaches (Table 2).

Table 3 lists the average AUCs, sensitivity values, and specificity values calculated on the holdout test sets. Of all the methods employed, prediction analysis for microarrays (PAM) performed the best in terms of AUC, sensitivity and specificity combined.

Metabolomics enrichment analysis highlighted six metabolic pathways as significantly disturbed in the CP brain as compared with controls. These include folate metabolism, propanoate metabolism, androgen and estrogen metabolism, androstenedione metabolism, pterine metabolism, and steroid metabolism (Figure 2).

## 3. Discussion

To our knowledge, this is the first study to use targeted and quantitative metabolomics to biochemically profile post-mortem brain tissue from people who died from CP and compared them with age- and gender-matched controls. Our univariate analysis of the concentration data highlighted 27 metabolites to be significantly different concentrations between CP and control brains (Table 1). 

We achieved consistently good diagnostic performance (AUC > 0.80) using six different Machine Learning approaches. PAM analysis, following cross validation, yielded an AUC (95% CI) = 0.930 (0.8–1) with a sensitivity and specificity of 0.899 and 0.855, respectively. LR had the smallest AUC among all the algorithms used. This was probably due to its sensitivity and not being the most ideal method for nonlinear analysis. When we looked at all the variables used as predictors in all of the models, we identified glycerophospholipids (PC ae C44:5, PC ae C44:5, PC ae C44:6, 40:1, 40:6) and urea to be the common denominators. 

In our univariate analysis, we found that glutamate was included in the top significantly different metabolites in CP brains. Glutamic acid, known as a key molecule in cellular metabolism, is the most abundant fast excitatory neurotransmitter in the nervous system [44]. Glutamic acid is believed to be involved in cognitive functions such as learning and memory in the brain due to its function in synapsis [44]. In brain injury or disease, excess glutamate can accumulate outside the cells. This process causes calcium ions to enter cells, leading to neuronal damage and eventual cell death, known as excitotoxicity [45]. Excitotoxicity due to glutamate occurs as part of the ischemic cascade and is associated with stroke and diseases like amyotrophic lateral sclerosis, lathyrism, and Alzheimer’s disease [46,47,48]. A fundamental process that leads to perinatal brain damage with hypoxic-ischemic injury is believed to be the damage to neurons with excitotoxicity [49]. 

The potential sources of cellular glutamate available for release during ischemia include astrocytes, oligodendrocytes, axons, and cells from neighboring structures such as the choroid plexus. Of these sources, ischemic glutamate release from astrocytes has been well characterized in gray matter [50] as well as periventricular white matter which is a lesion associated with chronic neurologic morbidity, especially CP seen in premature neonates [51]. Moreover, in animal models, prenatal magnesium sulfate use had prevented local glutamate level elevation and neurologic impairment after an excitotoxic brain lesion [52]. This effect was more significant in males compared to females [52]. It is not surprising that our results supported the previous reports on the importance of glutamate metabolism in lesions associated with CP.

Glycerophospholipids or phosphoglycerides are the most significant metabolites identified in CP in our machine learning techniques. Glycerophospholipids function in signal induction and transport. They provide the precursors for prostanglandins and leukotrienes [53] for biological responses [54]. They are also involved in apoptosis, modulation of the activities of transporters, and membrane-bound enzymes [55,56,57]. Marked alterations in neural membrane glycerophospholipid composition have been reported to occur in neurological disorders such as Alzheimer’s disease, depression, and anxiety [54,58,59]. These alterations result in changes in membrane fluidity and permeability. These processes along with the accumulation of lipid peroxides and compromised energy metabolism may be responsible for the neurodegeneration observed in CP [60,61]. Umbilical cord metabolomic profiles in neonates with perinatal asphyxia who have substantial risk to develop CP showed significant alterations in amino acids, acylcarnitines, and glycerophospholipids [62] similar to our findings in the brain tissue of patients with CP.

Machine learning techniques also identified urea as a good predictive variable across all of our models. In neurodegenerative disorders such as Huntington’s disease, changes in urea levels were identified in post-mortem brain tissues [63]. Widespread elevation of urea has also been reported in brain tissues with Alzheimer’s disease [64], suggesting that urea cycle disruption could also be a unifying pathogenic feature of neurodegenerative diseases. Excessive levels of urea and its nitrogenous precursor ammonia are neurotoxic, as evidenced by uremic encephalopathy and the urea cycle disorders. Urea cycle disorders are genetic disorders caused by a mutation that results in a deficiency of enzymes in the urea cycle [65]. These enzymes are responsible for removing ammonia from the blood stream. In urea cycle disorders, nitrogen accumulates, resulting in hyperammonemia that can cause irreversible brain damage, with manifestations ranging from lethargy and abnormal behavior such as disordered sleep and neurological posturing through to acute psychosis, seizure, coma, and death [66]. Similarly, uremic encephalopathy typically occurs in patients with renal failure, which can lead to symptoms ranging from mild fatigue and generalized weakness to seizure and coma [67]. There have been no previous reports showing an association between urea cycle abnormalities and CP. Argininemia, which is a rare urea cycle defect disorder, has been reported in a small case series of young children leading to progressive spastic tetraplegia, poor physical growth, and mental retardation with seizures mimicking CP [68]. Our study is the first showing altered urea concentration in the post-mortem CP brain tissue supporting previous studies about other neurological disorders.

The results of the pathway enrichment analysis highlighted folic acid metabolism as the most perturbed biochemical pathway. Methylation cycle and folate metabolism are important in neurotransmitter regulation, nerve myelination, and DNA synthesis. Thus, folate metabolites play a critical role in cognitive function and neuromuscular stability. A previous study showed a possible protective effect of prenatal folic acid supplementation on CP development [69]. There is evidence that children with CP show dysregulation of methylation capacity and folate metabolism despite adequate levels of folate and vitamin B12 [70]. Maintenance of methylation activity is crucial for RNA and DNA synthesis and subsequent growth and development as well as maintaining neurodevelopment. Interestingly, there is a cerebral folate deficiency syndrome described in children with developmental delay and deceleration of head growth, psychomotor retardation, and hypotonia. One-third of these children develop ataxia, spasticity, dyskinesia, speech difficulties, and seizures similar to children with CP [71]. In mouse models, folate deficiency has been demonstrated to decrease neurotransmitter acethylcholine activity, which in turn significantly decreases cognitive performance [72]. Furthermore, low serum folate concentrations were also found in patients with Alzheimer’s disease and dementia [73]. There is also evidence of the beneficiary effect of folate therapy on both EEG patterns and neuropsychological performance in patients with neuropathy and cerebral atrophy [74].

Additionally, our pathway enrichment analysis identified propionate metabolism as being significantly perturbed in CP brains. Propionate is the most common short-chain fatty acid produced by the human gut microbiota in response to indigestible carbohydrates such as fiber in the diet. Propionate and other short-chain fatty acids are produced in the body during normal cellular metabolism following enteric bacterial fermentation of dietary carbohydrates and proteins [75]. Propionate-producing enteric bacteria, including unique *Clostridial*, *Desulfovibrio*, and *Bacteriodetes* species, have been isolated from patients with regressive autism spectrum disorders [76,77]. Propionate is also present naturally in a variety of foods and is a common food preservative in refined wheat and dairy products. Under normal circumstances, these short-chain fatty acids are primarily metabolized in the liver. However, if there are genetic and/or acquired aberrations in metabolism [78], higher than normal levels of short-chain fatty acids can be present in the circulating blood, and can cross the gut–blood and blood–brain barriers. They can concentrate intracellularly, particularly in acidotic conditions, where they may have deleterious effects on brain development and function [79]. This could be important in the context of neurological disorders, since propionate is known to affect cell signaling, neurotransmitter synthesis and release, mitochondrial function/CoA sequestration, lipid metabolism, immune function, gap junction modulation, and gene expression [79,80,81,82,83,84], all of which have been implicated in a variety of neurological disorders including autism spectrum diseases [79,85]. Intracerebroventricular infusions with propionate produced short bouts of behavioral and electrophysiological effects, coupled with biochemical and neuropathological alterations in adult rats, consistent with those seen in autism disorder [86,87,88,89]. A recent study showed infusions with propionate or butyrate altered the brain acylcarnitine and phospholipid profiles [90], which are known to affect membrane fluidity, peroxisomal function, gap junction coupling capacity, signaling, and neuroinflammation [79], supporting our findings as earlier defined in CP brain tissues.

Finally, we found that the sex steroid metabolism pathway was significantly altered in the brain tissue of patients with CP. Although there is paucity of data on the effect of sex steroids in the development of CP, estradiol has been shown to have a dose-dependent protection on oxygen-induced apoptotic cell death in oligodendrocytes in animal models [83]. This may suggest a possible role for estrogens in the prevention of neonatal oxygen-induced white matter injury [91]. Although sex steroid levels were low for both genders after birth, our preliminary finding should be investigated more deeply to identify the correlation with the better survival rates of female premature babies compared to males [92]. Estrogen could be effective in modulating glutamate-induced neurotoxicity [85]. However, the mechanism underlying estrogen’s neuroprotective effect is not fully clarified [93]. Moreover, as previously mentioned, there may be a gender-specific neuroprotective effect of magnesium sulfate in the premature brain [52]. When plasma levels of androgens were analyzed in male subjects with autism compared to males with mental retardation and control subjects, androgenic hormone levels were not different among the groups, except that the DHEAS levels were higher in mentally retarded patients with CP compared to age-matched mentally retarded patients without CP or controls [94].

In our study, the number of cases and controls was small due to the difficulties in obtaining the post-mortem brain tissues from patients with CP. Clinical information for both cases and controls was limited. The age and gender for cases and controls were matched with the best available samples in the NIH NeuroBioBank. The biopsy specimens were obtained from the same but one anatomical location of each brain to be analyzed. 

## 4. Materials and Methods 

### 4.1. Tissue Samples

Only a limited number of specimens and tissue was available for this pilot study. Tissue samples (caudate nucleus) were obtained from post-mortem CP cases (*n* = 9) and age- and gender-matched control subjects (*n* = 11). Tissues were obtained from the Harvard University Tissue Resource Center, the University of Maryland Brain and Tissue Bank, and the University of Miami Miller School of Medicine, which are all Brain and Tissue Repositories of the NIH NeuroBioBank. This study was approved by the Beaumont Health System’s Human Investigation Committee (HIC No.: 2018-387). The methods were carried out in accordance with the approved guidelines. Details such as age, gender, race, and post-mortem delay can be found in Appendix A in the Appendix A.

### 4.2. Sample Preparation

Samples were stored at −80 °C prior to preparation. Subsequently, samples were lyophilized and milled to a fine powder under liquid nitrogen to limit the amount of heat production. For ^1^H NMR, 50 mg samples were extracted in 50% methanol/water (1 g/mL) in a sterile 2 mL Eppendorf tube. The samples were mixed for 20 min and sonicated for 20 min, and the protein was removed by centrifugation at 13,000× *g* at 4 °C for 30 min. Supernatants were collected, dried under vacuum using a Savant DNA SpeedVac (Thermo Scientific, Waltham, MA USA), and reconstituted in 285 μL of 50 mM potassium phosphate buffer (pH 7.0), 30 μL of sodium 2,2-dimethyl-2-silapentane-5-sulfonate (DSS), and 35 μL of D_2_O [95]. A 200 μL portion of the reconstituted sample was transferred to a 3 mm Bruker NMR tube for analysis. All samples were housed at 4 °C in a thermostatically controlled SampleJet autosampler (Bruker-Biospin, Billerica, MA, USA) and heated to room temperature over 3 minutes prior to analysis by NMR.

For analysis by targeted mass spectrometry, the tissue samples were analyzed using the commercially available AbsoluteIDQ p180 (Biocrates, Innsbruck, Austria) kit. In brief, 10 mg (±3 mg) of milled tissue were extracted in 300 µL of solvent (85% ethanol and 15% phosphate buffered saline solution). The samples were shaken at 700 rpm for 10 min, followed by sonication for 20 min, and centrifuged at 13,000× *g* for 20 min. The supernatant was collected and 10 µL were used for analysis with the kit. A 10 µL portion of blank, 3 zero samples, 7 calibration standards, and 3 quality control samples were loaded onto the filters in the upper 96-well plate and dried under nitrogen using a positive pressure processor (Waters Technologies Corporation, Milford, MA, USA). Subsequently, 50 µL of phenylisothiocyanate derivatization solution were added to each well and left at room temperature for 20 min. The plate was subsequently dried under nitrogen for 60 min, followed by the addition of 300 µL of methanol containing 5 mM ammonium acetate for the extraction of metabolites. The plate was shaken at 700 rpm for 30 min and the extracts filtered to the lower collection plate using the positive pressure processor. Eluates were diluted with water for the analysis of the metabolites with the workflow using ultra-pressure liquid chromatography mass spectrometry (UPLC-MS) and diluted with running solvent for flow injection analysis (FIA)-MS (for lipids).

### 4.3. Data Collection and Metabolic Profiling

Using a randomized running order, all 1D ^1^H NMR data were recorded at 300 (±0.5) K on a Bruker ASCEND HD 600 MHz spectrometer (Bruker-Biospin, Billerica, MA, USA) coupled with a 5 mm TCI cryoprobe. For each sample, 256 transients were collected as 64k data points with a spectral width of 12 kHz (20 ppm), using a pulse sequence called CPP WaterSupp (Bruker pulse program: pusenoesypr1d) developed by Mercier et al. [96] and an inter-pulse delay of 9.65 s. The data collection protocol included a 180-s temperature equilibration period, fast 3D shimming using the z-axis profile of the ^2^H NMR solvent signal, receiver gain adjustment, and acquisition. The free induction decay signal was zero filled to 128k and exponentially multiplied with a 0.1 Hz line broadening factor. The zero and first order phase constants were manually optimized after Fourier transformation and a polynomial baseline correction of the free induction decay (FID; degree 5) was applied for precise quantitation. All spectra were processed and analyzed using Chenomx NMR Suite (v8.0, Chenomx, Edmonton, AB, Canada).

As previously noted, targeted MS analysis was carried out using AbsoluteIDQ p180 kit (Biocrates Life Sciences AG, Innsbruck, Austria). Data was acquired using a Xevo TQ-S mass spectrometer coupled to an Acquity I Class UPLC system (Waters Technologies Corporation, Milford, MA, USA) as per the manufacturer’s instructions. The system allows for the accurate quantification of up to 188 endogenous metabolites including amino acids, acylcarnitines, biogenic amines, glycerophospholipids, sphingolipids, and sugars. Sample registration and the automated calculation of metabolite concentrations and export of data were carried out with Biocrates MetIDQ software. We accurately identified and quantified 59 metabolites using ^1^H NMR and 173 using DI/LC-MS/MS. Some overlap was observed (22 metabolites) between the two platforms and as such, we reported the average values for each individual metabolite measured using both analytical platforms.

### 4.4. Statistical Analysis

Using MetaboAnalyst (v4.0) [97], the data were analyzed using a two-tailed Student’s *t*-test to determine the statistical significance between the metabolite concentration in CP and corresponding controls (*p* < 0.05, FDR < 0.3).

We selected a representative set of six artificial intelligence algorithms, which have been applied for problems of data classification in the bioinformatics field. These included logistic regression (LR), prediction analysis for microarrays (PAM), partial least square-discriminant analysis (PLS-DA), deep learning (DL), random forest (RF), and support vector machine (SVM).

Using publicly available toolboxes in R, important parameters for each model were optimized so that the best prediction performance could be achieved [98,99,100,101,102,103]. In order to assess model performance of each approach or algorithm, the data were split into training and testing sets (80% and 20% respectively). In an attempt of avoiding sampling bias, the splitting process was repeated ten times and the AUC values were averaged out. Sensitivity and specificity values were calculated at 95% confidence intervals.

### 4.5. Metabolite Pathway Enrichment Analysis

Metabolite set enrichment analysis (MSEA) was completed using MetaboAnalyst (v4.0) [97]. Metabolite names were converted to Human Metabolome Database (HMDB) identifiers. The raw data was subjected to sum normalization and autoscaling. The pathway-associated metabolite set was the chosen metabolite library, and all compounds in this library were used. Pathways with a raw *p* value < 0.01 were considered to be significantly altered upon CP. 

## 5. Conclusions

We report for the first time a targeted, quantitative metabolomic approach for profiling post-mortem human brain tissue from patients with CP. Metabolomic analysis provided new insights into the dysregulated brain metabolism associated with CP. The metabolites and associated biochemical pathways identified herein could potentially facilitate the understanding of the underlying complex pathophysiology associated with CP as well as possible central biomarkers for early detection and prediction of CP. There is a need for future studies to confirm our current preliminary data in more accessible biomatrices.

## Figures and Tables

**Figure 1 metabolites-09-00027-f001:**
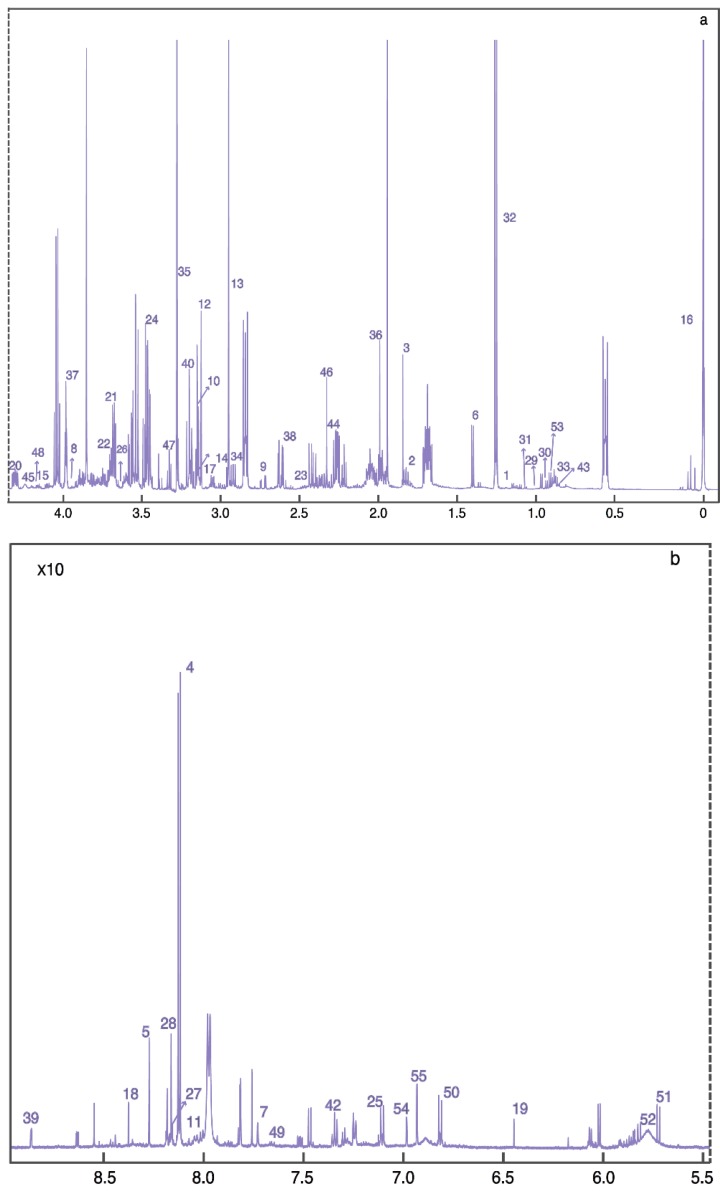
Typical (**a**) aliphatic and (**b**) aromatic region of 600 MHz ^1^H-NMR spectra of brain tissue extract, the metabolites are listed as follows. 1: 3-Hydroxybutyrate; 2: 4-Aminobutyrate; 3: Acetate; 4: Adenine; 5: Adenosine; 6: Alanine; 7: Anserine; 8: Ascorbate; 9: Aspartate; 10: Carnitine; 11: Carnosine; 12: Choline; 13: Creatine; 14: Creatine phosphate; 15: Creatinine; 16: DSS; 17: Ethanolamine; 18: Formate; 19: Fumarate; 20: Glucose; 21: Glutamate; 22: Glutamine; 23: Glutathione; 24: Glycine; 25: Histamine; 26: Homocitrulline; 27: Hypoxanthine; 28: Inosine; 29: Isobutyrate; 30: Isoleucine; 31: Isopropanol; 32: Lactate; 33: Leucine; 34: Lysine; 35: Methanol; 36: Methionine; 37: Myo-inositol; 38: N-Acetylaspartate; 39: Niacinamide; 40: O-Acetylcholine; 41: O-Phosphocholine; 42: Phenylalanine; 43: Propylene glycol; 44: Pyruvate; 45: sn-Glycero-3-phosphocholine; 46: Succinate; 47: Taurine; 48: Threonine; 49: Tryptophan; 50: Tyrosine; 51: Uracil; 52: Urea; 53: Valine; 54: π-Methylhistidine; 55: τ-Methylhistidine.

**Figure 2 metabolites-09-00027-f002:**
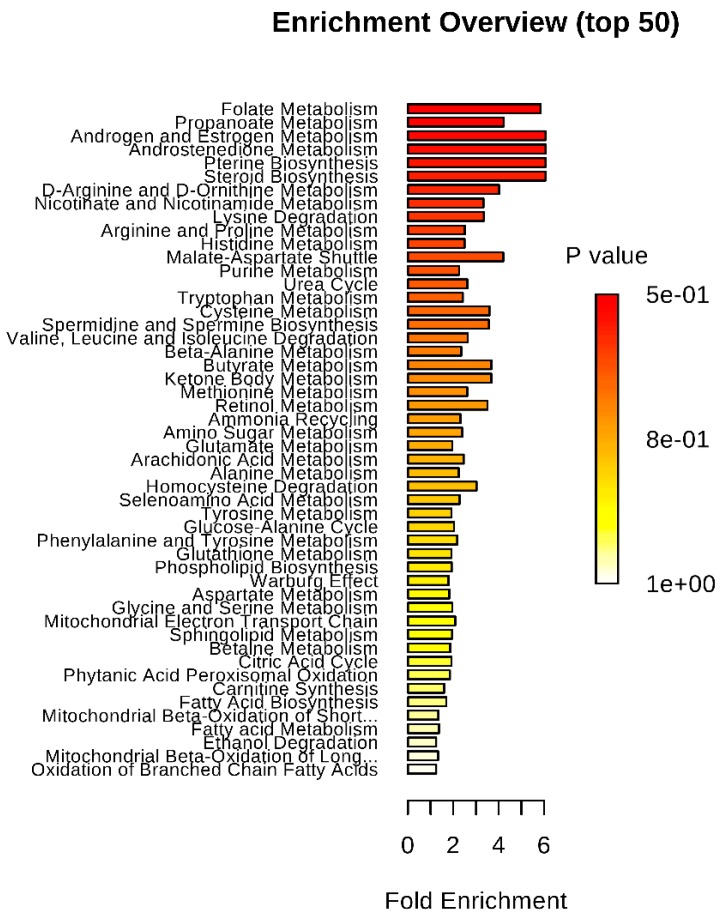
Results of the metabolite pathway enrichment analysis.

**Table 1 metabolites-09-00027-t001:** Statistically significant metabolite concentrations (μM; *p* < 0.05; *q* < 0.05) for CP vs control PM brain extracts. *t*-test values were calculated as a default and values with (W) were calculated using the Wilcoxon–Mann–Whitney test.

HMDB	Compound ID	Mean (SD) of Control (μM)	Mean (SD) of CP (μM)	*p*-Value	*q*-Value (FDR)	Fold Change
HMDB00294	Urea	59.236 (37.499)	184.144 (14.774)	0.0074 (W)	0.299	−3.11
HMDB00148	L-Glutamic acid	499.627 (15.680)	6.767 (13.764)	0.0106 (W)	0.299	73.83
HMDB13456	PC(o-22:2(13Z,16Z)/22:3(10Z,13Z,16Z))	1.187 (0.902)	0.335 (0.379)	0.0125 (W)	0.299	3.54
HMDB08276	PC(20:0/20:2(11Z,14Z))	0.265 (0.190)	0.051 (0.110)	0.0166 (W)	0.299	5.16
HMDB13450	PC(o-22:0/22:6(4Z,7Z,10Z,13Z,16Z,19Z))	0.847 (0.710)	0.231 (0.404)	0.0166 (W)	0.299	3.66
HMDB00195	Inosine	8.082 (4.627)	14.333 (6.338)	0.0201	0.299	−1.77
HMDB13333	3-Hydroxy-9-hexadecenoylcarnitine	0.061 (0.062)	0.129 (0.076)	0.0204 (W)	0.299	-2.13
HMDB10379	LysoPC(14:0)	5.237 (1.153)	4.151 (0.665)	0.0224	0.299	1.26
HMDB13433	PC(o-18:1(9Z)/22:0)	1.334 (0.714)	0.638 (0.487)	0.023	0.299	2.09
HMDB13453	PC(o-22:1(13Z)/22:3(10Z,13Z,16Z))	0.281 (0.180)	0.133 (0.069)	0.0248	0.299	2.12
HMDB07991	PC(16:0/22:6(4Z,7Z,10Z,13Z,16Z,19Z))	55.251 (4.352)	19.532 (5.971)	0.0249	0.299	2.83
HMDB08055	PC(18:0/22:5(4Z,7Z,10Z,13Z,16Z))	9.151 (6.281)	3.871 (2.773)	0.0249	0.299	2.36
HMDB06083	Troxerutin	188.555 (18.953)	432.889 (25.759)	0.0250 (W)	0.299	−2.3
HMDB08048	PC(18:0/20:4(5Z,8Z,11Z,14Z))	114.082 (59.935)	56.311 (43.130)	0.0264	0.299	2.03
HMDB00142	Formic acid	4.718 (2.078)	7.489 (3.055)	0.0269	0.299	−1.59
HMDB08057	PC(18:0/22:6(4Z,7Z,10Z,13Z,16Z,19Z))	23.314 (15.829)	11.438 (6.380)	0.0275 (W)	0.299	2.04
HMDB07892	PC(14:0/22:6(4Z,7Z,10Z,13Z,16Z,19Z))	0.405 (0.338)	0.139 (0.090)	0.028	0.299	2.91
HMDB0029205	LysoPC(26:0)	0.227 (0.197)	0.456 (0.235)	0.0293	0.299	−2.01
HMDB07874	PC(14:0/18:2(9Z,12Z))	3.462 (3.478)	0.558 (0.715)	0.0297 (W)	0.299	6.21
HMDB03334	Symmetric dimethylarginine	0.638 (0.399)	1.405 (0.802)	0.0310 (W)	0.299	−2.2
HMDB10394	LysoPC(20:3(8Z,11Z,14Z))	1.213 (0.902)	0.492 (0.500)	0.0310 (W)	0.299	2.46
HMDB08288	PC(20:0/22:6(4Z,7Z,10Z,13Z,16Z,19Z))	0.367 (0.230)	0.186 (0.100)	0.0332	0.299	1.98
HMDB11151	PC(O-16:0/18:2(9Z,12Z))	10.915 (6.853)	5.759 (2.592)	0.0381	0.299	1.9
HMDB13469	SM(d18:0/24:1(15Z)(OH))	1.353 (0.764)	2.168 (1.131)	0.0402 (W)	0.299	−1.6
HMDB13458	PC(o-24:0/18:3(6Z,9Z,12Z))	0.909 (0.441)	0.536 (0.290)	0.0428	0.299	1.7
HMDB08138	PC(18:2(9Z,12Z)/18:2(9Z,12Z))	189.522 (12.500)	60.640 (6.755)	0.0465 (W)	0.299	3.13
HMDB13411	PC(o-16:1(9Z)/16:1(9Z))	0.720 (0.496)	0.362 (0.212)	0.048	0.299	1.99

**Table 2 metabolites-09-00027-t002:** List of panels of metabolites used in different artificial intelligence methods. LR: logistic regression; SVM: support vector machine; PLS-DA: partial least square-discriminant analysis, RF: random forest; PAM: prediction analysis for microarrays; DL: deep learning.

Models	Selected Features
LR	PC ae C44:5, Urea
SVM	PC ae C44:5, Urea, C9
PLS-DA	PC ae C44:5, Urea, C9, PC aa C40:6, PC ae C40:1, PC ae C44:6
RF	PC ae C44:5, Urea, C9, PC aa C40:6, PC ae C40:1
PAM	Urea, PC ae C44:5, PC ae C44:6, C9, PC aa C40:6, PC ae C40:1
DL	C9, PC ae C40:1, Urea, PC ae C44:6, PC ae C44:5

**Table 3 metabolites-09-00027-t003:** Results for the various predictive modeling techniques employed.

	LR	SVM	PLS-DA	RF	PAM	DL
AUC (95% CI)	0.861 (0.688–1)	0.925 (0.73–1)	0.929 (0.8–1)	0.899 (0.6–1)	0.93 (0.8–1)	0.937 (0.8–1)
Sensitivity	0.842	0.778	0.870	0.889	0.899	0.833
Specificity	0.909	0.625	0.725	0.850	0.855	0.667

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
