# Peer review of "Metabolomic Profiling of Cerebral Palsy Brain Tissue Reveals Novel Central Biomarkers and Biochemical Pathways Associated with the Disease: A Pilot Study"

_metabolites, 2019, doi:10.3390/metabo9020027_

Round 1

Reviewer 1 Report

Metabolites-433631

Current submission ‘Metabolomic profiling of cerebral palsy brain tissue reveals novel central biomarkers and biochemical pathways associated with the disease: a pilot study’ by Savasan et al. presents expansion of research for prognosis of Cerebral Palcy (CP).

CP is a developmental disorder that causes disability in young children. It is usually caused due to brain injury either during fetal development or birth. Understanding pathophysiology is important for gaining insights into disease, there by developing preventive and protective strategies. Predictable knowledge of CP is very limited; towards this the authors aimed to utilize underlying potential of metabolomics to understand biochemical identity to CP. For the first time, post-mortem brain tissues from patients of CP were used for metabolomics study. Sample size of 20 cadnate nucleus tissue samples for both CP and control subjects were used for this study. Utilizing 1D 1H NMR and DI-LC-MS/MS methods they identified nearly ~ 200 metabolites and found 27 metabolites to be significantly varying in CP and healthy controls. Among these 27 metabolites, authors further narrow it down to metabolites glycerophospholipids and urea for discriminating CP vs control. They also identify folate, propanoate and androgen/estrogen pathways to be most perturbed in CP.

Authors discuss elaborately these select metabolites which are helpful in discriminating CP and control and those pathways that are perturbed. They also relate the perturbation of these metabolites in other neurological disorders like ASD, Alzheimer’s, dementia etc. This study demonstrates the potential of metabolomics in understanding biochemistry and to predict/identify CP in patients is quite encouraging and it promises greater chances of prediction of CP in children.

Overall, the manuscript is well written, literature citation is comprehensive and covers all important citations, Tables, figures and figure legends are consistent with the presentation. Material and methods well described and easy for peers to understand and replicate the study. Therefore, I feel this manuscript will be of good interest to the researchers in the community.

Minor points that should be addressed:

1.       Discussion should be shortened. Description of formation /pathway of metabolites should be reduced. For example, urea generation need not be discussed. At the same time urea recycling in case of genetical mutations related urea metabolism is discussed but evidence or lack of evidence of these mutations in CP is not discussed

2.       Few additional citations that involve using of post-mortem brain tissue for metabolomics study of neurological disorder should be included.

3.       Additional figure of representative 1D 1H NMR spectrum with labeled metabolites should be included

4.       Table-1 metabolites list is extensive. It should be limited only to significant 27 metabolites and the rest should be moved to supplementary materials

5.       Line 157 LR performed worst of all the algorithms. This statement should be rephrased

6.       Line 317: our study has limitation. This line can be deleted. Limitation in sample size can be understood in the exploratory studies. Repetitive mention of limited sample size can be avoided.

7.       Line 416: it should be ‘Metabolite set enrichment analysis

Reviewer 2 Report

Attached.
